# Comprehensive Analysis of the Yield and Leaf Quality of Fresh Tea (*Camellia sinensis* cv. Jin Xuan) under Different Nitrogen Fertilization Levels

**DOI:** 10.3390/foods13132091

**Published:** 2024-07-01

**Authors:** Jiajun Cai, Zihao Qiu, Jinmei Liao, Ansheng Li, Jiahao Chen, Zehui Wu, Waqar Khan, Binmei Sun, Shaoqun Liu, Peng Zheng

**Affiliations:** 1College of Horticulture, South China Agricultural University, Guangzhou 510642, China; 13600708173@163.com (J.C.); scau20222018004@stu.scau.edu.cn (Z.Q.); las1533854642@stu.scau.edu.cn (A.L.); cjhtea@stu.scau.edu.cn (J.C.); 18202000019@163.com (Z.W.); waqar.khan399@gmail.com (W.K.); binmei@scau.edu.cn (B.S.); scauok@scau.edu.cn (S.L.); 2Soil and Fertilizer Station of Cenxi City, Wuzhou 543200, China; ljm19127614165@stu.scau.edu.cn

**Keywords:** nitrogen levels, quality, yield, Jin Xuan, fresh tea leaves, gene expression

## Abstract

Reasonable application of nitrogen fertilizer can improve the yield and quality of tea. This study used Jin Xuan as the tested variety and applied nitrogen fertilizer at rates of 0 kg/ha (N0), 150 kg/ha (N150), 300 kg/ha (N300), and 450 kg/ha (N450) in the summer and autumn seasons to analyze the effects of nitrogen application on the quality components and gene expression of tea leaves. The results showed that the N150 treatment significantly increased total polyphenols (TP), total catechins (TC), and caffeine contents, with the most significant increase observed in the content of six monomers of catechins (EGCG, ECG, EGC, GCG, GC, and EC) in the summer. The N300 treatment significantly increased TP and AA contents in the autumn while decreasing TC content. Additionally, the N300 treatment significantly increased caffeine and theanine contents in the autumn. Notably, the N300 treatment significantly increased both summer and autumn tea yields. Multivariate statistical analysis showed that TPs, AAs, TCs, EGC, and caffeine were key factors affecting the quality of Jin Xuan. Furthermore, the N150 treatment upregulated the expression of the phenylalanine ammonia-lyase (*PAL*) gene, which may increase the accumulation of catechins. In conclusion, it is recommended to apply 150 kg/ha of nitrogen fertilizer in the summer and 300 kg/ha of nitrogen fertilizer in the autumn. This recommendation provides a theoretical basis for improving the quality and yield of tea leaves in summer and autumn.

## 1. Introduction

Nitrogen is one of the most important constant nutritional elements in plant growth and development [1], and it is also the most scarce mineral element in soil [2], participating in the metabolic pathways of many important quality components of tea, such as tea polyphenols, free amino acids (AAs), and caffeine [3]. Nitrogen application can enhance soil nutrient supply, thereby maintaining the normal physiological metabolism of tea plants, which is crucial for preserving tea quality and yield [4]. For example, increasing the supply of nitrogen fertilizer increased the concentration of amino acids in tea and decreased the concentration of polyphenols [5]. Research conducted in southern India indicated a correlation between higher levels of nitrogen fertilizer application and increased tea yield [5]. A study found that reasonable nitrogen fertilization (200–350 kg/ha) can significantly increase tea yield and affect the quality-related components of fresh tea leaves, such as tea polyphenols, AAs, caffeine, and volatile aroma compounds [6], which are known to influence the flavor quality of tea significantly. Zhang et al. [7] also found that nitrogen application at a rate of 300–450 kg/ha can ensure the nitrogen requirement for the normal growth of tea plants. Insufficient nitrogen fertilization can lead to stunted growth, premature senescence, and decreased yield in tea plants [8]. Conversely, excessive nitrogen application can reduce the plant’s nutrient uptake capacity, making it difficult to improve both the yield and the quality of tea leaves [9], and even lead to a series of adverse environmental issues, such as soil acidification, soil degradation, and water pollution [10]. Therefore, optimizing nitrogen supply measures is significant for increasing both tea yield and quality.

The tea plant (*Camellia sinensis* L.) is one of the most important economic crops in the world, playing an important role in the economies of many countries [11]. Jin Xuan (*Camellia sinensis* cv. Jin Xuan) is a suitable cultivar for oolong tea developed by the Tea Research Institute of Taiwan in China, with excellent agronomic traits, such as high yield, strong adaptability, and resistance. Due to the varying biochemical composition and content of fresh tea leaves of different qualities, the quality of raw leaves significantly influences the taste, aroma, and nutritional properties of tea [12]. The abundant biochemical components in tea determine its unique flavor and quality [13]. The content and ratio of catechins, AAs, and caffeine are key factors in determining the taste and quality of tea. Tea polyphenols, as the most important secondary metabolites in tea, determine the bitterness and astringency of tea infusions [14]. AAs play a crucial role in the formation of the fresh and sweet taste of tea infusions [15]. Caffeine, as an important nitrogen-containing compound, primarily determines the bitterness of tea infusions and is approximately twice as bitter as EGCG [16]. Tang et al. [17] found that nitrogen fertilizer application significantly affects the content of TPs, AAs, and caffeine in tea leaves. Ruan et al. [18] also found that adequate nitrogen fertilizer application increases the content of AAs and chlorophyll in tea leaves and reduces the content of TPs, thereby lowering the TP/AA ratio and enhancing the freshness and mellowness of green tea flavor. Previous studies have indicated that nitrogen levels can affect the production of primary and secondary metabolites, thereby influencing the flavor of tea [19]. Furthermore, under different nitrogen levels, the carbon–nitrogen allocation ratio in tea plants differs significantly, which significantly impacts the physiological metabolic cycle within the tea plant [18]. The phenylalanine ammonia-lyase (*PAL*) gene is a key enzyme involved in the initial step of phenylalanine biosynthesis of flavonoids, regulating the primary metabolic flux of carbon into this pathway [14]. A study by Wang et al. [14] indicated that the expression of PAL in shoots with one bud and two leaves without nitrogen treatment was significantly higher than in other nitrogen treatments. The initial assimilation process of plant nitrogen involves the conversion of ammonium nitrogen into glutamine through the *GS/GOGAT* cycle, which then generates various amino acids for protein biosynthesis [20]. Wang et al. [14] found that nitrogen supplied in the form of NO3^−^ and NH4^+^ significantly upregulates the expression of *GOGAT*. However, the comprehensive impact of nitrogen levels on tea leaf quality components and related gene expression in different seasons remains unclear. Therefore, with the trend of efficient management in tea gardens, it is necessary to investigate the response of tea leaf quality components and related gene expression to nitrogen levels in different seasons.

This study identified and quantified the biochemical components related to tea quality from field experiments, analyzed the changes in gene expression, and employed multivariate statistical analysis to identify the most influential biochemical components in order to systematically analyze the relationship between nitrogen levels and the quality of fresh Jin Xuan tea leaves. Our findings clarified the nutrient requirements of tea leaves for nitrogen in different seasons, provided new insights into how nitrogen affects tea quality, and offered theoretical references for improving nitrogen utilization efficiency in tea gardens as well as tea quality.

## 2. Materials and Methods

### 2.1. Chemicals and Reagents

Caffeine standards (99.9%) were obtained from the Beijing Weiye Research Institute of Metrology and Technology (Beijing, China). Reference standards, including theanine (98%), catechin (C, 98%), epicatechin (EC, 98%), gallocatechin (GC, 98%), epigallocatechin (EGC, 98%), epicatechin gallate (ECG, 98%), gallocatechin gallate (GCG, 98%), and epigallocatechin gallate (EGCG, 98%), were purchased from Shanghai Yuanye BioTechnology Co., Ltd. (Shanghai, China). Ultrapure water (type 1) was generated using a Barnstead GenPure Pro system (Thermo Fisher Scientific, Waltham, MA, USA).

### 2.2. Experimental Design and Collection of Tea Samples

The tea plantation for this experiment is located at the South China Agricultural University in Guangzhou, Guangdong Province, China (23.16° N, 113.36° E). The average temperature of the tea garden in summer is 26–32 °C, and the average temperature in autumn is 23.0–31.0 °C. The tea plant variety selected was Jin Xuan. Urea served as the experimental fertilizer and met the requirements of GB/T 2440-2017 [21] for agricultural urea. The mass fraction of N in urea is greater than or equal to 45.0%. The experiment employed a randomized complete block design, with each plot covering an area of 1.2 square meters, spaced 1.5 m apart, and each plot had an average of 6 tea trees. Four treatments were applied: N0 (0 kg/ha), N150 (150 kg/ha), N300 (300 kg/ha), and N450 (450 kg/ha), with three replications. Fertilizer application was performed by trenching: a trench 10 cm deep and 15 cm wide was opened below the leaf drip line, and the fertilizer was applied and covered with soil. Fertilizer application timing and proportions were as follows: 30% at the end of April 2022, 20% at the end of June 2022, 20% at the end of August 2022, and 30% at the end of October 2022. Sampling was conducted in summer (8 June 2022) and autumn (1 October 2022) using sampling frames measuring 33 cm × 33 cm, with the sampling standard being one bud and two leaves. Samples were rapidly frozen in liquid nitrogen and stored in a −80 °C ultra-low-temperature freezer. Random sampling was performed for all leaves, with three biological replicates for each sample group.

### 2.3. Determination of Total Polyphenols

Total polyphenol (TP) contents were determined by spectrophotometry following the national standard GB/T 8313-2018 [22]. A quantity of 0.20 g of dry tea powder was mixed with 20 mL of 70% methanol aqueous solution, manually shaken, extracted in a 70 °C water bath for 10 min, filtered hot with double-layer filter paper, and made up to 50 mL with distilled water. Then, 1 mL of the extract solution was taken, mixed with 4 mL of ultrapure water, 5 mL of ferric tartrate solution, and 15 mL of pH 8.0 buffer solution (20.32 mg/mL disodium hydrogen phosphate dodecahydrate and 1.39 mg/mL potassium dihydrogen phosphate ultrapure water). Absorbance was measured at a wavelength of 760 nm using a UV spectrophotometer (Shimadzu Instruments, Suzhou, China), and the TP content was calculated using a standard curve (Appendix A). The analysis was repeated three times for average values, and dry weight (DW) was used in the calculations.

### 2.4. Determination of Free Amino Acids

The determination of free amino acid (AA) contents was conducted following the national standard GB/T 8314-2013 [23]. A quantity of 0.20 g of dry tea powder was thoroughly mixed with 20 mL of 70% methanol aqueous solution, extracted in a 70 °C water bath for 10 min, filtered hot with double-layer filter paper, and then adjusted to 50 mL with distilled water. Then, 1 mL of the extract solution was taken and mixed with 0.5 mL of pH 8.0 buffer solution (22.71 mg/mL disodium hydrogen phosphate dodecahydrate and 0.46 mg/mL potassium dihydrogen phosphate in ultrapure water) and 0.5 mL 2% ninhydrin (diluted with 0.8 mg/mL stannous chloride in ultrapure water). The absorbance at the 570 nm wavelength was measured using a UV spectrophotometer (Shimadzu Instruments, Suzhou, China), and the free amino acid content was calculated using a standard curve (Appendix A). The analysis was repeated three times for average values, and DW was used in the calculations.

### 2.5. Determination of Chlorophyll

The determination of chlorophyll contents followed the previous method [24]. Exactly 0.5 g of fresh leaves were weighed into a test tube, and leaf segments were cut into 2 mm × 1 cm fragments. The leaf segments were immersed in acetone–95% ethanol (1:2) under dark conditions until the leaves turned completely white. The absorbance was measured at 645 nm and 663 nm wavelengths using a UV spectrophotometer (U-5100, Hitachi, Tokyo, Japan). The content of chlorophyll a and b was calculated using the Arnon formula. The analysis was repeated three times for average values, and DW was used in the calculations.

### 2.6. Analysis of Caffeine, Theanine, and Catechins

The contents of caffeine, theanine, and catechins were analyzed using HPLC (Waters Alliance 2695, 2489 UV/Vis; Waters Technologies, Milford, MA, USA) to compare their retention times with standard substances. Calibration curves were used for quantitative analysis [25]. The content of caffeine, theanine, and catechins was calculated using calibration curves (Appendix A). The analysis was repeated three times for average values, and DW was used in the calculations.

Caffeine extraction was performed by mixing 0.1 g of freeze-dried tea powder with 30 mL of 1.5% magnesium oxide solution, followed by extraction in 100 °C ultrapure water for 30 min. After centrifugation at 13,000× *g* for 10 min, 1 mL of the supernatant was collected through a 0.22 µm microporous membrane filter. Then, 10 µL of the filtrate was injected into an XSelect HSS C18 SB column (4.6 × 250 mm, 5 µm; Waters Technologies, Milford, MA, USA) at a column temperature of 35 ± 1 °C and a flow rate of 0.9 mL/min. The mobile phase consisted of 100% methanol (A) and 70% ultrapure water (B), detected at a wavelength of 280 nm.

Theanine extraction was performed by immersing 0.1 g of freeze-dried tea powder in 10 mL ultrapure water at 100 °C for 30 min, followed by filtration of the supernatant through a 0.22 µm microporous membrane. Then, 10 µL of the filtrate was injected into an RP-C18 column (250 mm × 4.0 mm, 5 µm, 35 ± 1 °C) at a flow rate of 0.5 mL/min. The specific HPLC method referred to previous research [25], with the detection wavelength set at 210 nm.

Catechin extraction was performed by sonicating 0.20 g of freeze-dried tea powder in 8 mL of 70% methanol (Xin Zhi Biotechnology Co., Ltd., Ningbo, China) for 30 min, followed by centrifugation at 13,000 rpm for 10 min. The supernatant (1 mL) was filtered through a 0.22 µm microporous membrane and injected into an XSelect HSS C18 SB column (4.6 × 250 mm, 5 mm; Waters Technologies, Milford, MA, USA) maintained at a temperature of 25 ± 5 °C. A gradient elution method was employed with a mobile phase of 0.1% formic acid aqueous solution (A) and 100% acetonitrile (B). The gradient elution started at 8% B, increased to 25% from 5 to 14 min, and then decreased to 8% from 14 to 30 min, with the detection wavelength set at 280 nm.

### 2.7. Gene Expression Analysis by qPCR

To conduct gene expression analysis, fresh tea leaves were ground with liquid nitrogen. Then, the Magen plant total RNA extraction kit was used to extract sample RNA, and the RNA concentration was determined using a nucleic acid analyzer. The integrity of the extracted RNA samples was assessed by 1% agarose gel electrophoresis, and they were stored at −80℃ for future use. The target gene sequences were synthesized by Guangdong QIANGKE Biotech Company (Guangzhou, China) (Appendix A). The RNA template was reverse transcribed into cDNA templates using the EZBioscience reverse transcription kit (Guangdong QIANGKE Biotech Company, Guangzhou, China). The reaction systems were 10 μL, repeated three times for each system, considering pipetting errors; the total volume of the mixed system generally needs to reach 35 μL (including 16 μL of the reaction mix, 16 μL of ddH2O, 0.5 μL of forward and reverse primers, and 3 μL of cDNA template). The qRT-PCR experiments were performed using the Roche LightCycler 480 fluorescence quantitative PCR instrument according to the PCR reaction program (Appendix A), and the relative gene expression analysis was performed according to the 2^(Actin-A)^ method.

### 2.8. Statistical Analysis

The means and standard deviations of biochemical component, yield, and gene expression values from three replicates were calculated using Microsoft Excel 2021. GraphPad Prism 8.0 software (GraphPad Software, Inc., La Jolla, CA, USA) was utilized for one-way analysis of variance (ANOVA) and Duncan’s multiple comparison test was used to assess the significance of differences among the samples. Meanwhile, the main biochemical component and gene expression data were analyzed. Samples were considered to have significant differences when *p* < 0.05. SIMCA-P 14.1 was employed for principal component analysis (PCA), partial least-squares discriminant analysis (PLS-DA), and variable importance projection (VIP).

## 3. Results and Discussion

### 3.1. Effects of Summer Nitrogen Application on Yield and Biochemical Components

#### 3.1.1. Effect of Summer Nitrogen Application on Yield

The rate of increase following nitrogen application is calculated as the difference between each fertilization treatment and N0, divided by N0. The average rate of the N150, N300, and N450 treatments represents the overall rate. The application of appropriate nitrogen fertilizer can promote the accumulation of nutrients in tea leaves, thereby significantly increasing the yield of tea leaves in tea cultivation [26]. This study found that nitrogen fertilization significantly increased tea yield in summer, indicating a promoting effect of nitrogen on tea tree growth (Figure 1A). Compared to the N0 treatment, the yield increases under the N150, N300, and N450 treatments were 9.9%, 35.0%, and 15.4%, respectively, with an average increase of 20.1%. The yield under the N300 treatment showed a significant difference compared to the N0 treatment, with a yield of 55.43 kg/ha, consistent with the experimental results of Qiu et al. [27]. A previous study found that the application of synthetic nitrogen fertilizer increased tea yield by nearly 70% [28].

Furthermore, the results indicate a non-linear relationship between nitrogen levels and tea yield, with overall yield showing an initial increase followed by a decrease. Both excessive and insufficient nitrogen application can have adverse effects on tea yield. Excessive nitrogen application leads to soil pH exceeding the suitable range for tea tree growth, affecting normal growth and resulting in reduced tea yield [29]. Insufficient nitrogen application results in nutrient deficiency during tea tree growth, affecting shoot development and hindering yield improvement. In countries with higher average tea yields, such as India, Sri Lanka, and Kenya, nitrogen fertilizer application generally does not exceed 300 kg/ha [30], which aligns with our results.

#### 3.1.2. Effect of Summer Nitrogen Application on Chlorophyll

In order to assess the effects of different nitrogen levels on the quality of tea leaves, this study analyzed the variations in 13 major biochemical components known to have varying degrees of impact on leave quality in fresh tea under different nitrogen levels (Appendix A).

Most of the nitrogen is usually allocated to chloroplasts to participate in photosynthesis in the process of nitrogen metabolism in plants, and the content of photosynthetic pigments determines the intensity of photosynthesis [31]. Chlorophyll, an important photosynthetic pigment in tea leaves, causes an unpleasant grassy smell in excessive quantities [32]. In this study, compared to the N0 treatment, nitrogen application increased the total chlorophyll (Chla + Chlb) content in summer tea leaves with an average increase of 6.6%, the Chla + Chlb content increased from 1.37 mg/g to 1.46 mg/g (Figure 1B). The content of Chla and Chlb increased by 3.6% and 12.0%, respectively. The Chla content increased from 1.02 mg/g to 1.06 mg/g, and the Chlb content increased from 0.36 mg/g to 0.40 mg/g.

The N450 treatment showed the most significant increase in the content of Chla and Chlb among the nitrogen application treatments, with increases of 12.7% and 41.7%, respectively. In contrast, the chlorophyll ratio (Chla/Chlb) significantly decreased under the N450 treatment, with a decrease of 21%, and the differences between the other treatments were not significant, indicating that the chlorophyll structure of tea leaves grown under moderate to low nitrogen conditions is relatively stable. These results indicate that the N450 treatment is most effective in promoting photosynthesis in summer tea leaves, but it may cause the tea to have a grassy aroma and affect the flavor.

#### 3.1.3. The Influence of Nitrogen Application in Summer on the Total Polyphenol Content and the Total Content of Free Amino Acids

In this study, nitrogen fertilization significantly influenced the content of TPs and AAs during summer (Figure 1A). Compared to the N0 treatment, nitrogen fertilization increased the TP content by 7.2% and decreased the AA content by 8.2%, thereby increasing the TP/AA ratio. The TP content increased from 269.48 mg/g to 288.88 mg/g, and the AA content increased from 34.52 mg/g to 37.35 mg/g. Consistent with previous research, TP and AA contents exhibited opposite trends [33].

It was found that tea polyphenols have beneficial effects, such as antioxidant activity, effectively promoting health [34]. There were significant differences among the nitrogen treatments compared to the N0 treatment. The N150 treatment exhibited the most significant enhancement in TP content, with an increase of 19.3%. With the increase in nitrogen levels, TP content showed an overall trend of first increasing and then decreasing, consistent with the results of Ruan et al. [18].

The freshness of tea infusions is mainly related to the content of AAs [35]. Compared to the N0 treatment, all nitrogen treatments reduced the AA content of tea leaves, with the greatest decrease observed in the N150 treatment at 12.5%, with no significant difference between the N300 and N450 treatments. Meanwhile, AA content initially decreased and then increased with the increase in nitrogen levels, contrary to the trend observed for TP content. TPs and AAs in tea leaves can represent carbon and nitrogen sources, respectively, resulting in significant differences in the carbon–nitrogen allocation ratio of tea plants under different nitrogen application rates [18]. However, in a conflicting study, the synthesis of AAs in hydroponically grown tea plants was minimal under conditions of nitrogen deprivation [36]. Additionally, Qiu et al. [27] used Lingtou Dancong (*Camellia sinensis* cv. Lingtou Dancong) as the experimental variety and found that nitrogen application effectively increased the total content of free amino acids in summer tea leaves and could improve the freshness of tea infusions, indicating that different tea varieties may exhibit significant differences in response to nitrogen fertilization. Therefore, further research is needed to investigate the relationship between nitrogen application levels and AA content in tea leaves.

The ratio of TPs to AAs (the TP/AA ratio) is one of the important factors affecting tea quality, with a lower ratio enhancing the freshness of tea infusions [14]. Compared to the N0 treatment, the N150, N300, and N450 treatments all increased the TP/AA ratio (Figure 1A) by 36.2%, 4.7%, and 10.5%, respectively, with the N150 treatment showing the greatest increase. The TP/AA ratio increased by an average of 17.2% in summer, indicating that nitrogen application may reduce the freshness of summer tea infusions and negatively impact tea flavor. In summary, the N150 treatment had the most significant impact on summer TPs, AAs, and TP/AA ratios.

#### 3.1.4. Effect of Summer Nitrogen Application on Caffeine, Theanine, and Catechins

Caffeine, catechins, and theanine contribute significantly to the flavor quality of tea [37]. In summer, nitrogen application reduced the levels of caffeine and theanine by 5.2% and 7.5%, respectively, while increasing total catechins (TCs) by 3.0%. Caffeine content decreased from 30.30 mg/g to 28.73 mg/g, and theanine content decreased from 3.30 mg/g to 3.05 mg/g, while TC content increased from 129.92 mg/g to 133.82 mg/g. Specifically, the N150 treatment showed the most significant increase in caffeine and TC contents (Figure 1C).

There were significant differences in caffeine content among nitrogen treatments, showing an overall trend of initially increasing and then decreasing, with the N150 treatment reaching the highest value; further nitrogen application did not increase caffeine content. The reductions in caffeine content for the N300 and N450 treatments were 6.9% and 11.6%, respectively; the lower caffeine content may have balanced the bitterness of summer tea infusions. However, contrary to previous studies, the caffeine content in green tea increased significantly with the rate of nitrogen application [2]; this may have been due to variety, temperature, and other factors. Compared to the N0 treatment, the average content of theanine decreased by 7.5%. Overall, there was a trend of initial decrease followed by an increase; except for the N450 treatment, nitrogen application significantly reduced the content of theanine, with the largest decrease observed in the N150 treatment (21.2%). This is in contrast to the study by Qiu et al. [27], who found that nitrogen fertilization can effectively increase the theanine content of summer and autumn tea.

Catechins have strong antioxidant activity and can effectively reduce the risk of cardiovascular diseases [38]. Compared to the N0 treatment, the N150 treatment showed the most significant increase in TC content by 9.7%, while the difference between the N300 and N450 treatments was not significant. This study determined six catechin monomers, namely, epigallocatechin gallate (EGCG), epicatechin gallate (ECG), epigallocatechin (EGC), gallocatechin gallate (GCG), gallocatechin (GC), and epicatechin (EC). Compared to the N0 treatment, nitrogen application reduced the content of two catechin monomers (EGCG and ECG) and increased the content of four other catechin monomers (EC, ECG, GCG, and GC) (Figure 1C). With the increase in nitrogen levels, all six catechin monomers showed a trend of initial increase followed by a decrease, with the largest increase observed in the individual content of catechins under the N150 treatment.

#### 3.1.5. Multivariate Statistical Analysis of the Primary Biochemical Components of Tea Leaves in Summer

Principal component analysis (PCA) was performed on the main biochemical component data to visually understand the differences in the main biochemical components of tea leaves at different nitrogen levels and in different seasons. The PCA model extracted two principal components, PC1 and PC2, with variance contributions of 63% and 28.3%, respectively. The cumulative variance contribution reached 91.3%, indicating that selecting PC1 and PC2 for sample analysis had good reliability and revealed different characteristics between treatments at different nitrogen levels. Except for the N300 treatment, each nitrogen application treatment was significantly separated from the N0 treatment. The similarity between the N300 treatment and the N150 treatment was higher than that between the N300 treatment and the N450 treatment (Figure 2A1). A biplot (Figure 2A2) indicated that the contents of TPs, TCs, EGCG, GCG, and theanine were higher in the N150 and N450 treatments, while the content of AAs was higher in the N0 treatment. These results suggest a positive correlation between nitrogen levels and TP, TC, EGCG and theanine contents and a negative correlation with AA content.

To screen for potential key biochemical components, a supervised analysis method, partial least-squares discriminant analysis (PLS-DA), was used to amplify inter-group differences and minimize intra-group differences. The PLS-DA model confirmed the results of the PCA, showing a clear separation between different nitrogen levels (Figure 2B1). In this PLS-DA analysis, the fitting indices of the independent variables (R2X) and the dependent variables (R2Y) were 0.967 and 0.999, respectively. The model prediction index (Q2) was 0.998, and the intercept of the permutation test Q2 was less than 0, indicating that the model did not exhibit overfitting (Figure 2B2). The contribution of each variable to the PLS-DA model was represented by the variable importance in projection (VIP) (Appendix A), where variables with VIPs > 1 made significant statistical contributions. The results indicated (Figure 2B3) that TPs, EGC, caffeine, TCs, and AAs were identified as significant contributors, and these substances play essential roles in identifying tea leaves grown under varying nitrogen concentrations.

In summary, nitrogen application in summer significantly affects the yield and biochemical composition of tea leaves. Compared to the N0 treatment, the N150 treatment (150 kg/ha) significantly increased the content of TPs, caffeine, and TCs. Additionally, the N150 treatment had the most significant effect in terms of increasing the content of EGCG, ECG, EGC, GCG, GC, and EC. Ma et al. [33] also found that an annual average nitrogen application rate of 119–285 kg/ha is the best choice for increasing tea yield and quality. Nitrogen topdressing at a level of 105 kg/ha significantly increases the yield and quality of drip-irrigated tea leaves [35]. Therefore, this study suggests that applying 150 kg/ha (N150) of nitrogen fertilizer can effectively increase tea yield and overall quality in summer. Different nitrogen application rates have a significant impact on soil physical and chemical properties, and soil physical and chemical properties have a certain correlation with tea yield and quality. In future studies, we will add soil analysis of tea gardens.

### 3.2. Effects of Summer Nitrogen Application on Yield and Biochemical Components

#### 3.2.1. Effect of Autumn Nitrogen Application on Yield

In this study, nitrogen application significantly increased the yield of autumn tea leaves (Figure 3A), averaging 74.7%. Compared to the N0 treatment, the yield increments under the N150, N300, and N450 treatments were 59.3%, 103.7%, and 60.9%, respectively, with N300 showing the most significant increase.

However, the yield of autumn tea leaves did not linearly increase with nitrogen application. Instead, it showed a trend of initially increasing and then decreasing with the increase in nitrogen levels, consistent with the results of Tang et al. [17]. The yield reached its highest value under the N300 treatment, at 103.7 kg/ha, indicating that moderate nitrogen application can significantly increase tea yield, while excessive nitrogen application can curb tea yield [33].

#### 3.2.2. Effect of Autumn Nitrogen Application on Chlorophyll

Chlorophyll serves as the material basis for the color of tea leaves and tea infusions. It is also an important precursor for synthesizing aroma components, playing a crucial role in forming the flavor quality of tea leaves [39]. Compared to the N0 treatment, nitrogen application reduced the contents of Chla and Chlb in autumn tea leaves (Figure 3B) by 7.0% and 3.9%, respectively: Chla content decreased from 1.14 mg/g to 1.06 mg/g, and Chlb decreased from 0.51 mg/g to 0.49 mg/g. Specifically, the Chlb content significantly increased by 21.6% under the N150 treatment. A significant decrease in Chla/b was observed under the N300 treatment, with no significant differences observed in the other treatments. Chen et al. [26] also found that the highest chlorophyll content in tea leaves was observed with nitrogen application at 225 kg/ha, which promotes photosynthesis in tea leaves.

The increase in Chla + Chlb content was most significant under the N150 treatment, while further increases in nitrogen application led to a significant decrease. This suggests that moderate nitrogen application can promote chlorophyll synthesis, but chlorophyll content does not increase proportionally with nitrogen application, consistent with the findings of Qiu et al. [27].

#### 3.2.3. The Influence of Nitrogen Application in Autumn on the Total Polyphenol Content and the Total Content of Free Amino Acids

TPs and AAs have a positive impact on tea quality. Compared to the N0 treatment, nitrogen application treatment significantly increased the content of TPs and AAs in autumn tea by 14.5% and 16.1%, respectively. The TP content increased from 180.67 mg/g to 206.87 mg/g, and the AA content increased from 31.29 mg/g to 36.33 mg/g (Figure 3A), indicating a positive effect of nitrogen application on autumn tea quality. The N300 treatment showed the most significant increase in TP and AA contents in autumn tea, with 21.4% and 20.8% increments, respectively.

With the increase in nitrogen level, the TP content showed a trend of increasing and then decreasing (Figure 3A), the highest increment of 21.4% being observed for the N300 treatment. There was a significant decrease under the N450 treatment, consistent with the findings of Ruan et al. [18], who found a significant reduction in TP content in tea leaves with high nitrogen application. This likely resulted from excessive nitrogen fertilizer application, which enhances photosynthesis in tea plants, leading to more products being used for protein synthesis and limiting the conversion of some sugars to polyphenols, resulting in a decrease in TP content.

Compared to the N0 treatment, the N150, N300, and N450 treatments increased AA contents (Figure 3A) by 12.7%, 20.8%, and 14.7%, respectively, with no significant differences between nitrogen application treatments. Specifically, the highest AA content was observed in the N300 treatment. The results indicated that nitrogen application in autumn can effectively increase the AA content of tea leaves, enhancing the freshness and taste of tea infusions, consistent with the findings of Qiu et al. [27].

In autumn, there were no significant differences in TP/AA ratios among the treatments (Figure 3A). Only the N450 treatment reduced the TP/AA ratio by 6.6%, indicating that more fresh substances accumulated in tea leaves under the N450 treatment, which may improve the flavor quality of tea. Previous studies also found that nitrogen application not only increased AA contents but also reduced TP contents, thereby decreasing the TP/AA ratio in tea leaves [17,26].

#### 3.2.4. Effect of Autumn Nitrogen Application on Caffeine, Theanine, and Catechins

Caffeine, theanine, and catechins are important components that influence the flavor and aroma characteristics of tea leaves [29]. Nitrogen application significantly affects these biochemical components (Figure 3C). Compared to the N0 treatment, nitrogen application significantly increased caffeine and theanine contents by 7.1% and 26.7%, respectively, while reducing the TC content by 2.7%. The caffeine content increased from 27.60 mg/g to 29.56 mg/g, and theanine increased from 5.77 mg/g to 7.31 mg/g, while the TC content decreased from 109.97 mg/g to 107.00 mg/g.

Compared to the N0 treatment, the N150, N300, and N450 treatments significantly increased the caffeine content in autumn tea by 6.2%, 10.9%, and 4.3%, respectively (Figure 3C), with the N300 treatment showing the most significant increase. This is consistent with previous studies, where the caffeine content of tea plants grown for 14 months was lower under conditions of low nitrogen supply, and increased caffeine content was observed with higher nitrogen application levels [40]. Theanine is a unique amino acid in tea plants and is also one of the important components that contribute to the fresh and fragrant taste of tea [14]. Compared to the N0 treatment, all nitrogen application treatments significantly increased theanine contents (Figure 3C), with increases of 23.1%, 36.4%, and 20.8% for the N150, N300, and N450 treatments, respectively. The N300 treatment showed the most significant increase, and there were significant differences among the nitrogen application treatments, indicating that N300 treatment may be beneficial for theanine synthesis and accumulation and make tea taste fresh and fragrant to a certain extent. This is consistent with the study by Ran et al. [41], which showed that theanine content increased under conditions of application of moderate nitrogen fertilizer.

Catechins are the main components of tea polyphenols, accounting for about 70% of TP content [42]. Compared to the N0 treatment, nitrogen application significantly reduced the TC content of autumn tea leaves by 2.7%, with the most significant decrease observed in the N300 treatment, indicating that nitrogen application might balance the bitter effect of catechins. This is consistent with the findings of Pokharel et al. [40], indicating that moderate nitrogen application can significantly reduce the TC content of fresh leaves not affected by tea aphids. This study determined six catechin monomers and analyzed the relationship between different nitrogen levels and catechin monomers. Nitrogen application significantly affects the formation of catechins in fresh tea leaves, but its effects vary among different monomers [26]. The results indicated that nitrogen application reduced the content of five catechin monomers (EGCG, EGC, EC, GCG, and GC), ECG being the exception (Figure 3C). EC decreased by an average of 22.1%, and GCG decreased by an average of 33.7%. Among them, the N300 treatment significantly decreased the content of EGCG, EGC, EC, GCG, and GC compared to the N0 treatment. In conclusion, nitrogen application significantly affects the content of caffeine, theanine, and catechins in autumn tea leaves, significantly influencing the quality of autumn tea.

#### 3.2.5. Multivariate Statistical Analysis of the Primary Biochemical Components of Tea Leaves in Autumn

PCA was conducted on the major biochemical components of autumn tea leaves. The PCA model extracted two principal components, PC1 and PC2, with variance contribution rates of 59.2% and 26.7%, respectively, cumulatively explaining 85.9% of the total variance, indicating the reliability of selecting PC1 and PC2 for sample analysis. A clear separation between different nitrogen levels indicated distinct characteristics among nitrogen treatments (Figure 4A1). There were significant differences in biochemical component contents among nitrogen application treatments, with higher similarity between the N300 and N450 treatments than between the N300 and N150 treatments. A biplot (Figure 4A2) revealed higher levels of TPs, AAs, caffeine, and theanine under the N300 treatment; higher levels of Chlb, EGC, and GC under the N150 treatment; and higher levels of EGCG and GCG under the N0 treatment. These results suggest a positive correlation between nitrogen levels and the content of Chlb, EGC, AAs, TPs, theanine, and caffeine and a negative correlation with the content of EGCG and GCG.

PLS-DA was conducted on the major biochemical components of autumn tea leaves to investigate their contribution and abundance. The PLS-DA model confirmed the results of the PCA, showing clear separation between the varying nitrogen levels (Figure 4B1). In this PLS-DA analysis, the goodness-of-fit index (R2X) was 0.938, the goodness-of-prediction index (R2Y) was 0.986, and the model prediction index (Q2) was 0.947. The intercept of permutation test Q2 was less than 0, indicating no overfitting of the model (Figure 4B2). The contribution of each variable to the PLS-DA model was represented by the variable importance in projection (VIP) values (Appendix A). Variables with VIPs > 1, such as TPs, AAs, EGC, and TCs, were identified as important contributors (Figure 4B3), indicating that these components could serve as potential markers for distinguishing tea leaves grown under varying nitrogen concentrations.

In summary, compared to the N0 treatment, the N300 treatment (300 kg/ha) had the most significant improvement effect on TP, AA, caffeine, and theanine contents in autumn, with increases of 21.4%, 20.8%, 10.9%, and 36.4%, respectively, while reducing the TC content. It also significantly increased tea yield by 103.7%. Benti et al. [2] conducted nitrogen application treatments on Chinese tea plant varieties introduced in Ethiopia. They found that applying 300 kg/ha of nitrogen fertilizer could increase the TP content of tea leaves and reduce caffeine content, thereby benefiting the production of higher-quality tea. In addition, studies have shown that rational application of nitrogen fertilizer (200–350 kg/ha) can significantly increase the yield of tea and affect the quality components of fresh tea, such as free amino acids [43]. Therefore, this study suggests applying 300 kg/ha (N300) nitrogen fertilizer in autumn to effectively increase the yield and overall quality of autumn tea leaves.

### 3.3. The Effect of Nitrogen Application on Gene Expression Levels in Summer and Autumn

This study analyzed the effects of nitrogen levels on the expression of genes involved in the synthesis of major biochemical components in fresh tea leaves through qRT-PCR experiments, including genes related to catechin biosynthesis (*PAL*, *LAR*, and *DFR*), caffeine biosynthesis (*CsTCS1* and *CsSAM1*), theanine biosynthesis (*TCS1* and *CSTS1*), chlorophyll biosynthesis (*CsHEMA1* and *CsHEMA2*), and nitrogen metabolism biosynthesis (*GOGAT*). The varying nitrogen levels led to the upregulation and downregulation of the expression of the 10 genes studied (Figure 5A,B). In the summer, nitrogen fertilization upregulated the expression of six genes, with the highest upregulation observed in the N450 treatment. However, in the autumn, the N0 treatment upregulated the expression of seven genes, indicating significant seasonal differences in the effects of nitrogen application on the expression of genes related to biochemical component synthesis in tea leaves. These differences likely resulted from environmental factors such as temperature, light intensity, humidity, and soil nutrient levels.

Five genes (*PAL*, *LAR*, *GOGAT*, *CsHEMA1*, and *DFR*) exhibited opposite regulatory trends in summer and autumn, with increasing nitrogen levels (Figure 5A). In the summer, the expression levels of these genes initially increased and then decreased, while the opposite trend was observed in the autumn. However, the expression patterns of the remaining five genes (*CsTS1*, *CsTCS1*, *CsHEMA2*, *CsSAM1*, and *TCS1*) under nitrogen application were consistent in both summer and autumn seasons, with the expression levels of *CsTS1* and *CsHEMA2* initially increasing and then decreasing, while the expression levels of *CsSAM1* and *TCS1* initially decreased and then increased (Figure 5B). In this study, with increasing nitrogen application, the expression trends of genes related to the synthesis of quality components, such as theanine, caffeine, and chlorophyll, were inconsistent with the corresponding changes in biochemical component contents, indicating that these genes may not fully reflect the accumulation of biochemical components in fresh tea leaves.

Zhao et al. [44] found that the relative expression levels of the caffeine biosynthesis-related genes *CsSAM1* and *CsTCS1* did not show significant differences under conditions of soil nutrient deficiency compared to regular conditions, indicating that soil nutrient deficiency affects caffeine synthesis by regulating the availability of substrates and other factors rather than altering gene expression. Additionally, chlorophyll biosynthesis involves a series of enzyme-catalyzed processes encoded by genes [45,46]. Environmental factors such as light intensity, temperature, humidity, and soil nutrient levels can affect enzyme activity and gene expression. However, consistent with previous research [26], the expression of the *PAL* gene was upregulated under the N150 treatment compared to the N0 treatment. This expression trend coincides with the accumulation of catechins in fresh leaves, indicating its role in enhancing catechin content.

## 4. Conclusions

This study investigated the effects of different nitrogen levels on the key quality components and yield of fresh Jin Xuan leaves. The results showed that compared to the N0 treatment, the N150 treatment (150 kg/hm^2^) significantly increased the TP, TC, and caffeine contents in the summer by 19.3%, 9.7%, and 3.0%, respectively, while the enhancement effect on the content of six monomers of catechins (EGCG, ECG, EGC, GCG, GC, and EC) was most significant. The N300 treatment (300 kg/hm^2^) significantly increased the contents of TPs and AAs in the autumn by 21.4% and 20.8%, respectively, while reducing the total catechin (TC) content. Additionally, the N300 treatment significantly increased the contents of caffeine and theanine in the autumn by 10.9% and 36.4%, respectively. Notably, the N300 treatment significantly increased the tea yield, with increases of 35.0% in summer and 103.7% in autumn.

PCA and PLS-DA revealed clear separation between the varying nitrogen levels, with TPs, AAs, TCs, EGC, and caffeine contributing significantly in both summer and autumn. Furthermore, gene expression analysis indicated that the upregulation of the *PAL* gene under the N150 treatment might have increased the content of catechins in fresh tea leaves. In summary, applying 150 kg/hm^2^ of nitrogen fertilizer (N150) can effectively improve the quality of fresh Jin Xuan leaves in summer. Applying 300 kg/hm^2^ of nitrogen fertilizer (N300) has the best effect in terms of increasing the yield of fresh Jin Xuan leaves in summer. Additionally, applying 300 kg/hm^2^ of nitrogen fertilizer (N300) can effectively improve the yield and quality of fresh Jin Xuan leaves in autumn, achieving the optimal fertilization effect.

This study provides an important reference for a deeper understanding of the differences in the effects of varying nitrogen levels on the quality and related gene expression of fresh Jin Xuan leaves and for guiding rational fertilization practices in tea gardens. Sensory analysis can further validate the changes in tea quality. Furthermore, emphasizing the need to adhere to international regulations on fertilizer concentrations is essential for ensuring the reliability and applicability of our research findings in real-world tea cultivation practices. In future studies, we will combine sensory analysis to comprehensively analyze the integrated impact of nitrogen levels on tea quality.

## Figures and Tables

**Figure 1 foods-13-02091-f001:**
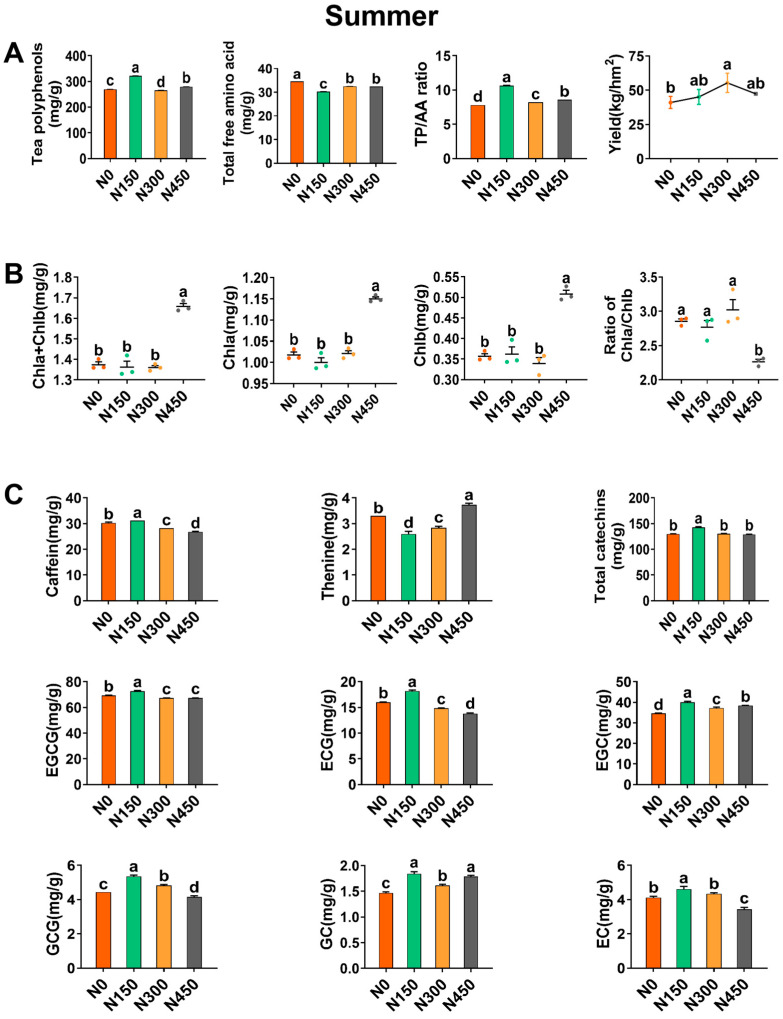
Analyses of the effects of varying nitrogen levels on the major biochemical components and yield of summer tea leaves. (**A**) Total polyphenols, total free amino acids, polyphenol–amino acid ratios, and yields at four nitrogen levels. (**B**) Content of chlorophyll a, chlorophyll b, chlorophyll a + b, and chlorophyll a/b. (**C**) The effect of nitrogen levels on the content of caffeine, theanine, total catechins, and monomeric catechins in tea leaves. The total catechin content was calculated as the sum of epigallocatechin gallate (EGCG), epicatechin gallate (ECG), epigallocatechin (EGC), gallocatechin gallate (GCG), gallocatechin (GC), and epicatechin (EC) contents. Data analysis was performed using one-way analysis of variance (ANOVA) and Duncan’s multiple comparisons. The analysis was repeated three times for average values, and DW was used in the calculations. All data are presented as mean values ± standard deviations. Different letters indicate a significant difference, with *p* < 0.05.

**Figure 2 foods-13-02091-f002:**
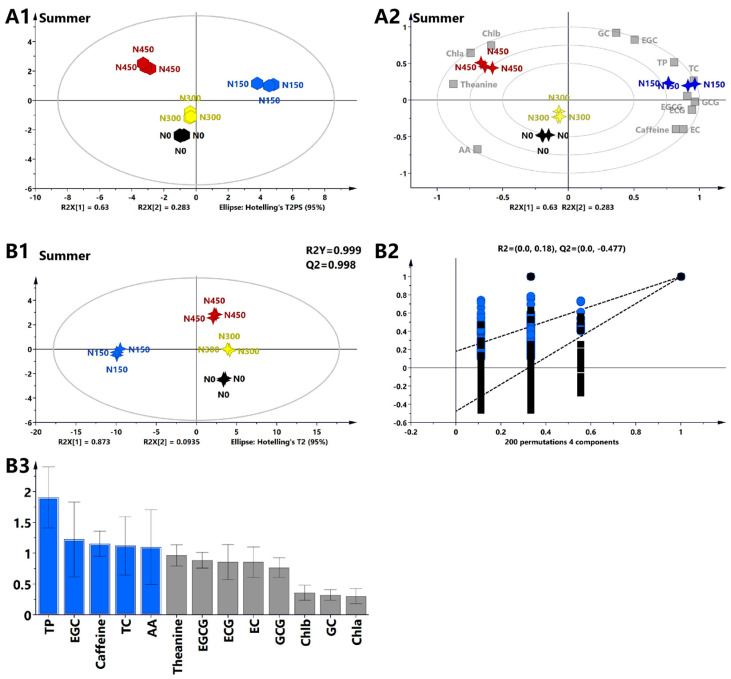
Analyses of the biochemical composition of summer tea under varying nitrogen levels using multivariate statistical analysis methods. (**A1**) PCA score plot showing the biochemical composition and quality parameters of summer nitrogen treatments, with R2X = 0.994 and Q2 = 0.973. (**A2**) Biplot revealing the correlation between four nitrogen levels in summer and 13 biochemical components. (**B1**) PLS-DA score plot showing varying nitrogen treatment levels. (**B2**) Cross-validation results indicating that after 200 permutations, the intercept of the Q2 regression line in the cross-validation model is less than 0. (**B3**) Variable importance in projection (VIP) results. Biochemical components with a VIP > 1 are represented by blue bars, while those with a VIP < 1 are represented by gray bars.

**Figure 3 foods-13-02091-f003:**
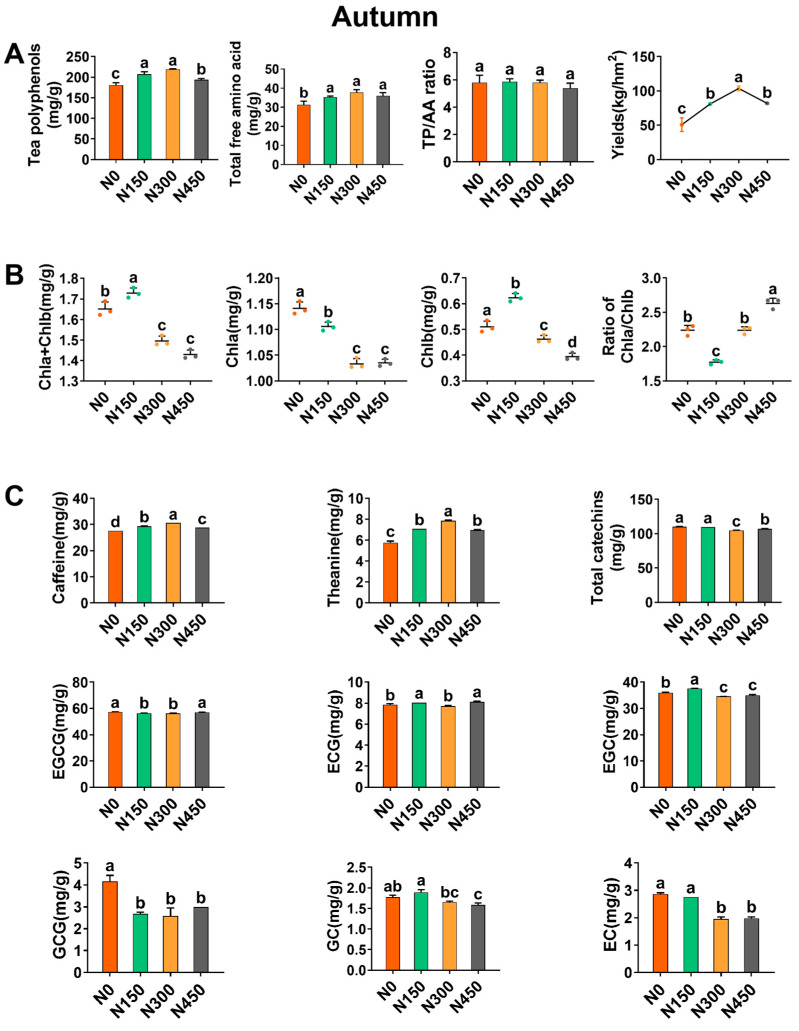
Analyses of the effects of different nitrogen levels on the major biochemical components and yield of autumn tea leaves. (**A**) Total polyphenols, total free amino acids, polyphenol–amino acid ratios, and yields at four nitrogen levels. (**B**) Content of chlorophyll a, chlorophyll b, chlorophyll a + b, and chlorophyll a/b. (**C**) The effect of nitrogen levels on the content of caffeine, theanine, total catechins, and monomeric catechins in tea leaves. The total catechin content was calculated as the sum of epigallocatechin gallate (EGCG), epicatechin gallate (ECG), epigallocatechin (EGC), gallocatechin gallate (GCG), gallocatechin (GC), and epicatechin (EC) contents. Data analysis was performed using one-way analysis of variance (ANOVA) and Duncan’s multiple comparisons. The analysis was repeated three times for average values, and DW was used in the calculations. All data are presented as mean values ± standard deviations. Different letters indicate a significant difference, with *p* < 0.05.

**Figure 4 foods-13-02091-f004:**
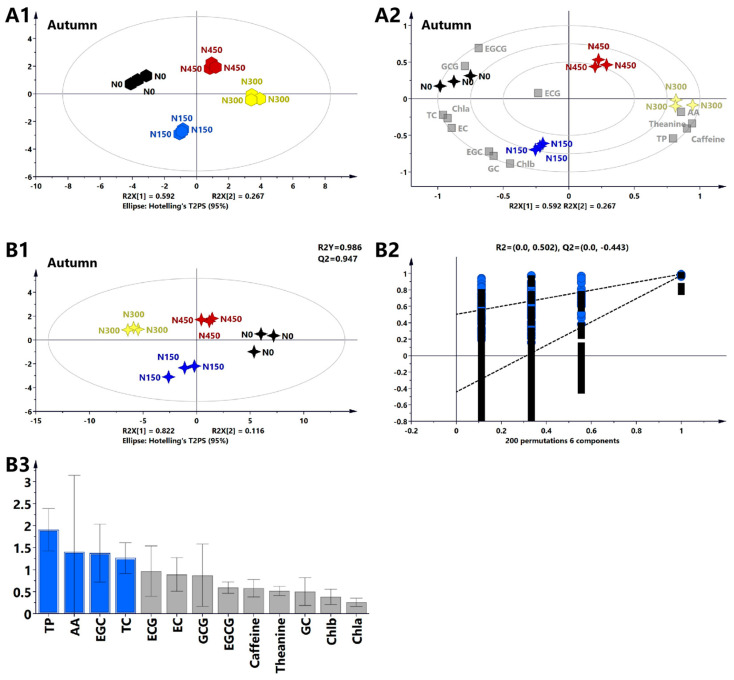
Analyses of the biochemical composition of autumn tea under varying nitrogen levels using multivariate statistical analysis methods. (**A1**) PCA score plot showing the biochemical composition and quality parameters of autumn nitrogen treatments, with R2X = 0.992 and Q2 = 0.909. (**A2**) Biplot revealing the correlation between four nitrogen levels in autumn and 13 biochemical components. (**B1**) PLS-DA score plot showing varying nitrogen treatment levels. (**B2**) Cross-validation results indicating that after 200 permutations, the intercept of the Q2 regression line in the cross-validation model is less than 0. (**B3**) Variable importance in projection (VIP) results. Biochemical components with a VIP > 1 are represented by blue bars, while those with a VIP < 1 are represented by gray bars.

**Figure 5 foods-13-02091-f005:**
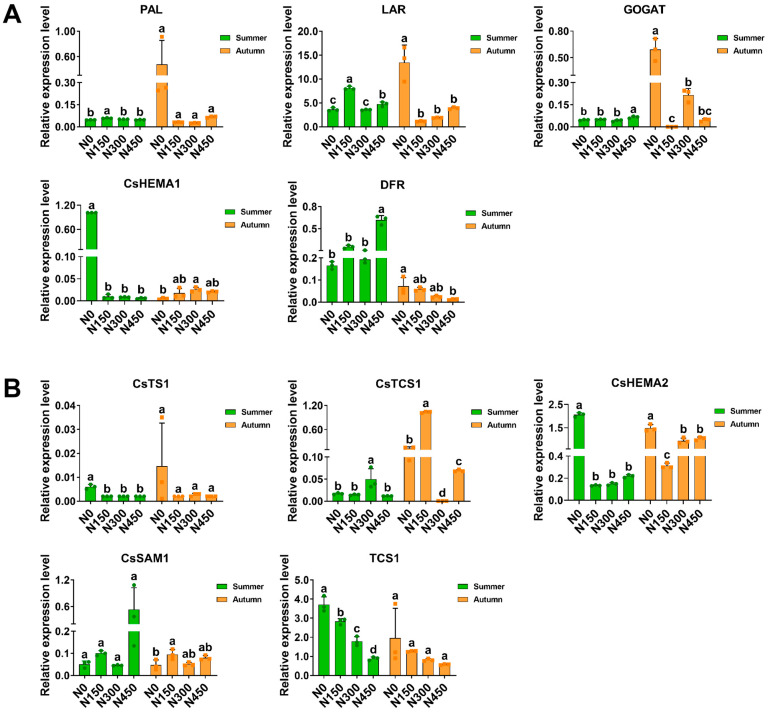
Analyses of the impacts of different nitrogen levels on the expression levels of genes related to the synthesis of major biochemical components in tea leaves during the summer and autumn seasons. (**A**) Five genes with opposing expression trends. (**B**) Five genes with similar expression trends. Data were evaluated using one-way ANOVA and subjected to Duncan’s multiple range test. Different letters indicate significant differences, with a significance level of *p* < 0.05.

## Data Availability

The original contributions presented in the study are included in the article/Appendix A; further inquiries can be directed to the corresponding author.

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
