# Peer review of "Comprehensive Analysis of the Yield and Leaf Quality of Fresh Tea (Camellia sinensis cv. Jin Xuan) under Different Nitrogen Fertilization Levels"

_foods, 2024, doi:10.3390/foods13132091_

Round 1

Reviewer 1 Report

Comments and Suggestions for Authors

This manuscript aims to study the effect of different nitrogen levels as biochemical components and gene expression in tea leaves grown in summer and autumn seasons. This article seems technically sound, however, there are many issues which should be clarified and improved prior to possible publication of this manuscript:

1.        In the experiment design, the tea plantation should be described in more detail. For instance, how many tea trees were grown? how many tea leaves were collected? what was the total weight of each treatment before and after drying? what was the temperature difference in summer and autumn?

2.        Instead of determination of total polyphenols, it would be better to determine total phenolic acids and total flavonoids separately.

3.        Instead of determination of free amino acids, it would be better to determine individual amino acid by HPLC to obtain a more reliable result.

4.        It would be better for the authors to determine CHL a and CHL b by HPLC for more accurate quantitation.

5.        The quantitation method of caffeine, theanine and catechin should be described in more detail. Did the authors use internal standard or external standard for quantitation? What was the standard concentration range? The quantitative data should be described in the result section and shown in a table including total polyphenol, free amino acid, CHL, caffeine, theanine and catechin in summer and autumn leaves to make it easy for comparison. The retention time data of all the bioactive compounds should be shown.

6.        P163-164: This statement is controversial.

7.        P160: “5 mm” should be corrected as “5 mm”.

8.        P158: What is “oxidase”?

9.        P156, 165 and 171: “quantification” should be corrected as “extraction”.

10.     Some more in-depth discussions should be provided. For example, why TP and AA contents exhibited opposite trends? Why the N150 treatment shown the most pronounced increase in TP content by 19.3% and the largest decrease in AA content by 12.5%? Why the N150 treatment showed the most significant increase in the contents of TC, caffeine and 6 catechin monomers? Why the 6 catechin monomers showed an initial increase followed by a decrease?

11.     Which one is more importantly, yield or content of bioactive compounds? It seems the N300 treatment produced the highest yield while the N150 treatment generated the highest levels of TP, caffeine, TC and 6 catechin monomers. This point should be clarified.

12.     The difference in contents in bioactive compounds including total polyphenol, free amino acids, CHL a, CHL b, caffeine, theanine and 6 catechin monomers between summer and autumn tea leaves should be compared carefully and explained. It would be better for the authors to show the contents of all the bioactive compounds in a table to make it easier for comparison.

Comments on the Quality of English Language

Minor editing of English language required

Reviewer 2 Report

Comments and Suggestions for Authors

The manuscript includes a practical study focused on the preparation of enriched tea leaves.

Title

Not clear: “at different N levels” this could be better expressed.

Abstract

Some global recommendation is missing. What is the recommended choice according to the results obtained ?

Material and methods

Some justification on the fertilizer concentrations checked ought to be provided. Are all of them allowed ?

Results

Sensory analysis was not carried out. The authors ought to refer on it and the possible effect of conditions tested.

Figure 1 and 3: According to the Material and methods section, the fact that triplicates (n = 3) were carried out ought to be indicated in the foot notes of the Figures.

Conclusions

The need for a complementary sensory analysis ought to be mentioned and carried out in on-coming research. Also, the need for compliance with international regulations for fertilizer concentrations ought to be expressed.

Comments on the Quality of English Language

Minor editing performances could be done.

Reviewer 3 Report

Comments and Suggestions for Authors

The introduction should end with a justification for the research being carried out. Authors should not include information about what they have tested in the experiment.

In the introduction, the authors sometimes refer to quite old literature sources - ten years old, and sometimes older. Please change it.

Both in the introduction and throughout the work, the authors refer almost exclusively to researchers from China. Tea appears to be grown and studied in other countries as well.

The method of determining the content of total polyphenols seems quite controversial. First of all, the use of a boiling water bath is questionable - polyphenols decompose at high temperatures. How does this translate into results? Moreover, a wavelength of 540 nm was used for measurements in this work. Could the authors explain which polyphenols absorb radiation at this wavelength? Standard determinations are made at 750 nm?

Polyphenols are very important compounds in tea that give it extraordinary antiradical properties. It seems that the profile of these compounds should be determined using a more advanced technique, e.g. HPLC or LCMS. This applies not only to catechins, but also to phenolic acids.

In the methodology, the analysis of caffeine and theanine should be separated from the determination of catechins.

The charts show units of mg/g. Please explain what the content of bioactive ingredients in tea is calculated into. Was dry matter determined at work?

Reviewer 4 Report

Comments and Suggestions for Authors

The manuscript is within the scope of the Journal. The English language is satisfactory, and readers will understand the information presented. The Authors aimed to identify and quantify the biochemical components related to tea quality by employing multivariate statistical. The Introduction, research design and method, results and conclusions have been adequately described. Statistical interpretations of the results have been provided. However, to maintain the impact and quality of the journal, it is required that the Authors revise their work according to the following comments:

1. Introduction: Lines 89-92 should be removed and placed appropriately in the Materials and Methods. 

2. Sections and subsections format/style should conform to section 2.6.

3. Text Paragraphs. Ensure that at least two to three paragraphs of the text or information under each section/subsection. Avoid too many paragraphs of the texts throughout the manuscript. Organize the information appropriately. 

4. Ensure clarity of the sentence in lines 202-203.

5. Lines 205-212 should be removed and placed appropriately in the Sections/Subsections of the results and discussion. 

6. Font size A, B, C and Summer in Figures 1 and 3 should conform to the required font size. 

7. Provide more references to support the results and discussion - Sections/subsections. The provided references are not adequate. For instance, in section 3.2.4, only two references were provided - [25], [37]; in section 3.2.5 has only one cited reference [39].

Reviewer 5 Report

Comments and Suggestions for Authors

Review

In this article, the authors used nitrogen fertilizer to improve the yield and quality of Camellia sinensis tea. They followed the influence of nitrogen rate supply on the total polyphenols, catechins, and caffeine contents during different seasons. They proved that N150 supply (nitrogen rate) upregulates the expression of phenylalanine ammonia-lyase. The subject area and smart use of fertilizer to obtain high-quality food and tea are important, and this article could be published, but in my opinion, there are some experimental data and explanations that must be included in order for the experiment to be reproducible and lead to correct conclusions. Therefore, it could be published only after including missing data.

Major Revision:

1.       Authors should also include rate supply or at least the content of potassium and phosphor since in agriculture, NPK fertilizers are often combined. Following only nitrogen without paying attention to P and K can lead to inconsistent results.

2.       In what form was N applied since there are also different types of NPK fertilizers or urea (it influences the pH of the soil)?

3.       Authors must emphasize the novelty of their research and whether there are other similar data on the influence of nitrogen rate supply on the quality of Camellia sinensis or similar teas. They should compare their results with literature data.

4.       Soil analysis must be included.

Minor Revisions:

1         There are typing errors in labeling anions (superscript) for example Line 79

Round 2

Reviewer 1 Report

Comments and Suggestions for Authors

The authors have satisfactorily addressed all the comments raised by reviewers and substantially improved the overall quality of the article. Therefore, I recommend accepting this article for publication in Foods.

Comments on the Quality of English Language

Minor editing of English language required

Reviewer 3 Report

Comments and Suggestions for Authors

Authors' explanation: "specific methods for the determination of bioactive ingredients of tea in this study are placed in the Materials and Methods section, and the content of these components are calculated by calibration curve, as detailed in Supplementary Information (Table S3). The dry matter content of tea has been determined by experiments. Thank you again for yours "careful review of our paper" is not sufficient. Of course, I understand that the results are converted from the standard curve, but I guess the results from the curve are not directly taken into account for interpretation, but only converted into content. Hence, I asked what specific games the result was converted into - we play fresh leaves, dry leaves or dry matter. Conversion to dry matter content allows comparison with other results from other researchers, while other conversions do not allow this.

Please explain how it happened that the parameters of the method for determining total polyphenols have undergone two fundamental changes, but the results of this determination have not changed at all???

Adding the formula each time: "Thank you again for your careful review of our paper." It is not necessary. Just say thank you once.

Reviewer 4 Report

Comments and Suggestions for Authors

The revised manuscript has potential for publication. However, the following comments should be considered for a minor revision:

1. The SI unit kg/ha appeared four times in lines 116-117 for the four treatments. But, the same treatments have different SI unit as kg/hm2 which appeared 24 times in entire the manuscript. I think the unit kg/hm2 is inappropriate whereas that of kg/ha is appropriate. If you agree, then correct the kg/hm2 throughout the manuscript to maintain uniformity of the SI unit for the four fertilizers/treatments.   

2. Section/Sub-section written style/format should conform to the Journal's Instructions for Authors. Please, refer to it. 

3. Reference list 43 should be checked. The title of the article/paper seems not to be correct. It seems a result was provided instead of the correct title of the paper. Indicate the name of the Journal in which the paper was published. Please, check also the title of ref. 44.

Reviewer 5 Report

Comments and Suggestions for Authors

I recommend the publication of the article, but after including a few sentences where they should add that in future studies, they will include soil analysis since it could influence their results (Comments 1 and 4).
